# The Earth as "Mother of Men" in Latter-Day Saint Theology

Kevin L. Tolley 

LDS Seminaries & Institutes of Religion, Riverside, CA 92507, USA; tolleykl@churchofjesuschrist.org

**Abstract:** Following the completion of work on the Book of Mormon, Joseph Smith began his work on expanding the Bible's scope. Unlike many of his contemporary Bible thinkers who were also working on translations of the Bible, Smith expanded the text in unique ways, breathing life into archaic and mysterious figures and developing themes far beyond the Biblical scope. Within the first year of the Church of Jesus Christ, Smith introduced significant information concerning a vision of the pseudepigraphical character of Enoch and additional information concerning the creation narrative. These additions give insight into Smith's understanding of his theology and his views on the environment. These additional writings connect environmental care and social injustice. The unique theological implication is that the treatment of the marginalized and downtrodden is closely related to the environment.

**Keywords:** mormon; latter-day saint; ecotheology; creation narrative; Enoch; Joseph Smith

## 1. Introduction

The theology of The Church of Jesus Christ of Latter-day Saints has stood apart from many denominations. The religion is framed as a restoration of ancient practices and covenants rather than perpetuating beliefs through the ages. Part of the unique character of the faith is the wealth of additional scripture that was produced by its founder, Joseph Smith. Besides a cadre of sermons and doctrinal discourses delivered over a fourteen-year career (1830–1844), Smith produced additional volumes of scripture along with additions to existing texts from the Bible. These additional texts give an expanded and unique perspective on human responsibility toward the earth. These extra-biblical texts become foundational to a Latter-day Saint perspective on the theology of environmental stewardship. However, among members of The Church of Jesus Christ of Latter-day Saints, Smith's teachings and distinct ecotheological perspective are under-utilized, and sometimes irrelevant to the daily religious practice of its members. Many are either unconcerned or ignorant of the ramifications of Smith's theological contribution concerning the contemporary environmental crisis. This paper will look at the Latter-day Saints' distinctive and insightful theological connection of human accountability toward the earth and their responsibility to care for the wellbeing of its inhabitants, especially the neglected and marginalized. This connection was instituted by Smith and other early church leaders and continues to expand and be taught today.

## 2. The Prophecy of Enoch

Following the completion of work on the Book of Mormon, Joseph Smith began his work on expanding the Bible's scope (Brown 2020). Unlike many of his contemporaries who were also working on translations of the Bible (Gutjahir 1999), Smith came to see such restoration of lost biblical texts as part of his prophetic mission. He grew the text in unique ways, breathing life into archaic and mysterious figures and developing themes far beyond the Biblical canon. Within the first year of the Church of Jesus Christ, Smith introduced significant information concerning the pseudepigraphical character of Enoch and an apocalyptic vision he had concerning the history of the world. The Bible contains only a brief mention of the ancient prophet. Genesis simply records, "Enoch walked

faithfully with God; then he was no more because God took him away" (Gen. 5:24). The Latter-day Saints established their first paper in Independence, Missouri, in June of 1832, called the Evening and Morning Star. The press was established in the printing office of W. W. Phelps. Among the first issues of the newspaper, Smith released the first sampling of his revised Biblical texts. He did not begin with his reworking of the creation fstory or the narrative of Eden, but with the "The prophecy of Enoch".

The prophecy of Enoch (Moses 7) sparked an understanding of and an excitement for doctrines that would eventually develop over Smith's prophetic career. It introduced doctrines that ranged from concepts of a premortal existence to the distinctive doctrine of theosis or divinization. Terryl Givens and Brian Hauglid wrote that while the Book of Mormon lent Smith "his indispensable aura of prophetic authority, the 'Prophecy of Enoch' provided a personal role model to inspire him and a blueprint to direct him. The Book of Mormon may be the keystone of Mormonism, but the prophecy of Enoch is its doctrinal foundation." (Givens and Hauglid 2019).

### 3. Vision of Mother Earth

The prophecy of Enoch covers the entire history of humankind, combining both Biblical themes and modern concerns. At one point in the vision, Enoch's prophetic view falls upon a personified depiction of Mother earth (cp. 2 Nephi 9:7; Mosiah 2:26; Mormon 6:15), weeping with sorrow.

The vision includes an outpouring of emotional responses. Within the context of the recorded vision, three distinct parties weep for the wickedness of humankind: God (Moses 7:28–29), the heavens (Moses 7:28, 37, 40), and Enoch himself (Moses 7:41, 49). In addition, the earth also laments; as the text records:

> And it came to pass that Enoch looked upon the earth; and he heard a voice from the bowels thereof, saying: Wo, wo is me, the mother of men; I am pained, I am weary, because of the wickedness of my children. When shall I rest, and be cleansed from the filthiness which is gone forth out of me? When will my Creator sanctify me, that I may rest, and righteousness for a season abide upon my face? (Moses 7:48)

It is not uncommon for biblical literature to personify ideas and concepts, many of which are embodied as female characters. Proverbs 8 envisions *wisdom* as a woman (Prov. 8:1, 4) who stood with God as the earth was framed (Prov. 8:22–29). The book of Jeremiah parallels other Sumerian–Akkadian texts by personifying significant cities as the mothers of their inhabitants (Jere. 10:19–21; Roberts 2002).

The text that Joseph Smith produced has remarkable parallels to other ancient texts. Scholars have noticed that the concept of a weeping earth appears in 1 Enoch (1 Enoch 7:4–6; 8:4; 9:2, 10; 87:1; Nickelsburg 2001) and in the *Book of Giants* found among the dead sea scrolls (4Q203, Frag. 8:6–12; Bradshaw 2014; Peterson 2002; Skinner 2002).

The Prophecy of Enoch uses figurative language to personify the earth. Although there are indications that the text itself is using metaphor, hyperbole, and symbolism, many early leaders saw the earth as alive and pushed the boundaries of animism. One of the first public Latter-day Saint sermons where the earth is declared to be alive was given by Orson Pratt in 1852. "The earth itself, as a living being, was immortal and eternal in its nature" (Pratt 1852).

The earth, envisioned as a lamenting mother who is pained over the wickedness of her children, takes up the role that Latter-day Saints often associate with a Heavenly Mother. The reaction of the earth parallels the reaction of God within the vision (Moses 7:28–29). The concept of a Heavenly Mother has subtly permeated Latter-day Saint theology since the early days of the Church. While there is no record of a formal revelation to Smith on the doctrine of a Heavenly Mother, the concept has remained (Paulsen and Pulido 2011; Petry 2020). Phelps wrote in 1844, "O Mormonism! Thy father is God, thy mother is the Queen of heaven" (Phelps 1844). Over 150 years later, Church leaders still refer to the theology of Heavenly Mother. Dallin H. Oaks claimed in 1997 that "our theology begins with heavenly parents" (Oaks 1995). Although there is no direct correlation between a Heavenly Mother

figure and the earth, they appear to share certain qualities; the concept of the earth being personified as a woman who weeps for her children might have been a catalyst, or at least a motivator, in the development of a female divine figure.

Enoch here sees the personified earth mourning over the flowing "filthiness" that is issuing from her children. The term "filthiness" refers to corruption, pollution, or defilement. The primary impetus behind the earth's plea for redress against the intense wickedness on her surface is to be cleansed and sanctified from the insidious wickedness. The cleansing appears to have two elements, cleansed from "wickedness" and cleansed from "filthiness." These parallel terms appear to be two sides of the same coin; the earth pleads for relief from pollution and sin. These twin elements are interwoven themes.

### 4. Additional Insights from Joseph Smith on Creation

Smith expanded the traditional scriptural canon, removing the bookends of the Bible. The Bible begins in Genesis with creation when God "created the heavens and the earth" and ends in the book of Revelation with the faithful being saved. Smith's additional scripture enlarged the view of the narrative by outlining what took place before creation and explains what occurs after salvation, expanding the view into the future and the past. As new scripture began to roll out among the Saints, the concept of a time before creation began to expand.

Smith continually returned back to the creation narratives. Latter-day Saint scripture repeatedly reviews the creation. While the Documentary hypothesis would later attribute the creation accounts in Genesis 1 and 2 to two separate authors, Smith viewed these two creation narratives as two different perspectives on the same event. He viewed it as a continuation of a singular process, as the actual creation of the physical world and its inhabitants (Genesis 1/Moses 2) as distinct from the record of an earlier creation (Genesis 2/Moses 3), which was a kind of spiritual prototype. He recorded that the earth was "spiritually before they were naturally upon the face of the Earth" (Smith 1831). The Latter-day Saint canon of scripture includes the Biblical record in Genesis 1–2, Smith's inspired commentary on the Biblical text (Moses 2–4), along with an additional version of creation included in a book entitled the Book of Abraham (Abraham 4–5). Smith felt so strongly about the creation narrative that he included it as part of the most sacred rites of the Church, the temple endowment. A review of the events of the creation has been a central feature of the most sacred of Latter-day Saint rites since the beginning of the endowment in Nauvoo.

In 1841, Smith was recorded as teaching, "God did not make the earth out of nothing for it is contrary to a rational mind and reason" (Smith 1841). Smith would unambiguously affirm eternalism in contrast to creation out of nothingness in 1842, when he declared "the elements are eternal" (Smith 1842). As opposed to ex nihilo, he emphasized ex materia. In the ancient near east, creation ex materia was common in polytheistic narratives such as in the Babylonian creation myth, *Enûma Eliš*. The concept of ex materia creation suggests that every act of creation must be understood as an act of re-creation. Weeks before Smith's death, he was recorded as interpreting the initial verses of Genesis. He said, "The word *create* came from the word baurau; it does not mean so; it means to organize" (Smith 1844). With this perspective, George Handley, a literary scholar and environmentalist, suggests that "God becomes less a creator or inventor and instead more an artist or a recycler," reusing and recreating new life from exisitng matter. Handley continues, "our stewardship must model his in this same way" (Handley 2020). Every new act of creation will, at minimum, impose itself as the partial loss of what came before. This struggle is illustrated within the seven days of creation (Gen. 1). The first three days are marked by elements of division and breaking apart; for instance, the light and darkness are torn apart on the first day, the firmament is separated on the second, and the waters are divided on the third. Each day is in preparation for the next series of days, days four through six, where the first days are filled with light and life, making possible further growth. Elements of chaos and darkness are replaced with luminaries and life. As opposed to "seeing this as an

oppositional struggle between order and disorder, we can see it as a kind of partnering that leads to life and diversity" (Handley 2020).

Smith emphasizes that God did not work alone in his creation. He utilized others in his work. The Book of Abraham records: "And then the Lord said: Let us go down. And they went down at the beginning, and they, that is the Gods, organized and formed the heavens and the earth" (Abraham 4:1). According to Smith, God utilized an entourage to assist in the work of creation. Building on the Biblical themes of a *Divine Council* that is periodically highlighted in Biblical text (*sôd*—Amos 3:7; Jeremiah 23:18), deity enlisted the help of others in the process of creation. This entourage was later identified as premortal children of God. Terryl Givens wrote, "Mormonism began to envision heaven as a place where God's own creative powers were mirrored in his exalted children" (Givens 2015). Parley P. Pratt, who was quick to embrace this new idea, wrote, "I might also tell you of the continued exertions of creative power by which millions of new worlds will yet be formed and peopled." He wrote of not only God the Father participating in this work but also "by King Adam and his descendants" (Pratt and Dye 2009). The Theology expanded even further. Not only were God's children involved with the creation process of the earth in a premortal state, but they will also, after leaving mortality, be involved in the process of creation again. This process of creating worlds was well established by 1841 (Givens 2015).

Not only were God's children involved in creating the earth, but, according to Susa Young Gates, the Heavenly Mother stood, "side by side with the divine Father, [was] the equal sharing of equal rights, privileges, and responsibilities" (Gates 1920). Jeffery and Patricia Holland also suggested that Heavenly parents were both involved in the ongoing process of creating everything and "are doing so lovingly and carefully and masterfully" (Holland and Holland 1989). In teaching the concept that God did not work alone, Smith suggests that God orchestrates and directs rather than mechanistically controls. The lesson is that the human family has a vested interest in the earth's wellbeing. Humanity becomes co-creators of a world that is never entirely independent. Humanity's influence on the earth continues. In this concept, God is directing the affairs of the world but delegates stewardship. Natural law governs the elements, but God's children are free agents and can choose to continue to follow the divinely appointed creative plan to bring light and life, or to destroy.

While working on a retranslation of Genesis in 1830, the following insertion was made into the text: "And every plant of the field before it was in the earth, and every herb of the field before it grew. For I, the Lord God, created all things, of which I have spoken, spiritually, before they were naturally upon the face of the earth. For I, the Lord God, had not caused it to rain upon the face of the earth" (Moses 3:5). Later, in 1843, Smith spoke on the issue of this spirit-matter: "there is no such thing as immaterial matter. All spirit is matter, but it is more fine or pure, and can only be discerned by purer eyes" (D&C 131:7). From these texts one gets a glimpse into his creation theology, which emphasizes the spiritual qualities of all of life. Smith taught of a pre-physical creation of "all things." Elements have some sort of reflection in a divine or spiritual realm. He did not suggest that the spiritual matter of nature was to be worshiped, but rather believed that there is a kinship between all living souls. Smith taught that there are binding links connecting humanity to the physical world. He taught the light of Christ, which enlightens and enlivens all creation (D&C 88:6–13). Latter-day Saint beliefs suggest a "relationship between the spirit and the body, the human role within creation, and social ethics reveals sufficient evidence that the restored gospel teaches that, as God's children, humans bear a heavy moral responsibility to act as stewards of all God's physical creations and to treat them with sustained respect and sustaining love" (Handley 2020). Throughout Smith's teachings, there is a consistent and pervasive theological implication that the spiritual realm is inseparably connected to the physical, that the one is reflected upon the other, and that damage to one will likewise be reflected upon the other.

Smith continued to reveal and review information concerning humanity's role in the creation of the earth. God did not act alone but was viewed as a co-worker within a divine

family. Involvement in the care for the earth continues as a divine mandate. This theology suggests that a spark of divinity runs through not only all humankind, but through all of God's creations. As part of this restoration, people continue to be stewards with a divinely commissioned responsibility to protect, represent, and foster that divine spark.

**5. Symbiotic Relationship with the Earth**

The responsibility to care for the earth continues. The wellbeing of humanity is inherently connected to the earth. As Smith continued to work through the Biblical text of Genesis, key elements of this symbiotic relationship become apparent.

Few doctrines held by The Church of Jesus Christ of Latter-day Saints are more distinct than their view of the Fall of Adam. Terryl Givens wrote that Smith's view of the Fall "was early and consistent" (Givens 2015). As opposed to viewing this event as the tragedy of human existence, it is framed in the light of a positive step forward. Unique to the restored conception of the fall is the idea that the fall was a choice and an opportunity to enable humankind to "have joy" (2 Nephi 2:25). Later, Smith recorded Eve as declaring "Were it not for our transgression we never should have seed, and never should have known good and evil, and the joy of our redemption" (Moses 5:11). Eve is heralded as a heroine as opposed to an ignorant accessory in the story. Other unique aspects flow from this distinct perspective. Since all living things have spiritual matter, they are also subject to the conditions of the fall and of redemption (D&C 29:23–26). Since nature continues to obey God's command, nature becomes oppositional to human endeavor because of human disobedience. As a result, both humankind and the earth were subject to the fall. This kinship with the earth is part of what makes it such a profound mirror. In Genesis as well, humanity is figured as a class of gardeners who "dress" and "keep" the land (Genesis 2:15). These terms are later associated with a Levitical priest's responsibility to care for the tabernacle (cp. Num. 3:7–8; Ex. 21:29, 36; Brown et al. 1906). They are to care for the divine sanctuary as divinely appointed representatives. Ancient priests who use these resources for personal gain are referred to as "scoundrels" (1 Samuel 2:12).

The care for the earth is interwoven with the destiny of humankind. Just as humankind is fallen, so too does the earth follow in a parallel state. It leaves nature ever subject to humankind's agency, bringing it into the realm of ethics by implying that, as George Handley suggests, we are "co-workers with God in the redemption of nature" (Handley 2020).

Joseph Smith's translation of the Genesis narrative did not end with the creation of the subsequent fall of humankind. Further information is given concerning the means and motive behind Cain's fratricide. Cain killed his brother in order to take his belongings. Cain also becomes the *Mahan*, a term that is defined as "the master of this great secret," which is that one "may murder and get gain" (Moses 5:31). The logical progression of this idea blooms and spreads like an insidious weed through almost every culture. People are a commodity, a dispensable resource to be taken advantage of. Variations of this "great secret" can be seen throughout history. Slavery, human trafficing, and other examples of extortion become reflections of the worst kept secret. The lives of those most easily utilized are the marginalized, the minorities, and those on the fringes of society.

Cain's divinely appointed punishment is meted out by the earth herself. Since Cain sowed violence through his brother's blood, the earth would react by not yielding "unto [him] her strength" (Moses 5:37). Smith's translation of the text emphasizes that nature itself would turn against Cain. Earth's reaction is instantaneous. The seed Cain planted would never grow, a problematic curse for one who was occupationally a farmer. Although it is clearly articulated in the Joseph Smith Translation of the Cain narrative, the theme of nature turning against the greed of humankind is also amply attested to in the Hebrew Bible. Although the Bible does not represent nature in any particular consistent way, among the prophetic writings, wickedness most often results in an uncooperative climate. As Micah says, "The land shall be desolate because of them that dwell therein, for the fruit of their doings" (Micah 7:13). Isaiah teaches that human and environmental health and

spiritual and physical wellbeing are interdependent and should therefore be mutually nurtured (Isaiah 35:1–7). William Dyrness wrote, "Morality, [man's] response to God and fertility of the earth are interrelated" (Dyrness 1987). This interdependence of human morality and creation means, as Hugh Nibley wrote, that "moral and physical cleanliness and or . . . pollution" are inseparable (Nibley 1990). There exists a symbiotic relationship between concern for the marginalized and concern for the creation. When greed overruns how one treats the marginalized and overpowers the need to help and lift, nature takes offense.

Building on this interconnection between creation and society, Smith's successor Brigham Young taught early pioneers this interdependence: "You are here commencing anew. The soil, the air, the water are all pure and healthy. Do not suffer them to become polluted with wickedness. Strive to preserve the elements from being contaminated by the filthy, wicked conduct and sayings of those who pervert the intelligence God has bestowed upon the human family" (Young 1860a). Young's apparent concern is with stewardship. Each settler was commissioned to improve the land, but the warning is apparent. The purity of the land is not limited to physical contaminants. Illicit behavior will corrupt the land as easily as toxins. This idea elevates environmental care to a spiritual endeavor. Spiritual impurity not only affects the soul of the sinner but also the earth itself (Moses 7:48).

### 6. The Ultimate Destiny of the Earth

The theology of the Church of Jesus Christ of Latter-day Saints includes the salvation of humanity and the eventual salvation of the earth. The earth mimics humanity's journey to exaltation. Embedded within Smith's Articles of Faith is the concept that Christ will eventually rule upon this planet after it has received its "paradisical glory" (Articles of Faith 10). The concept of the eventual salvation of the personified earth mimics the path necessary for humankind to follow. The earth began in an Edenic state (2 Nephi 2:22), then fell and became riddled with thorns and thistles (Genesis 3:18); however, the earth's ultimate fate is the site of the celestial kingdom.

In July 1830, Emma Smith was counseled to "make a selection of sacred hymns," (D&C 25:11). The first volume of hymns, containing ninety selections, was published in 1835. Many of these hymns were unique to the faith. Several talented Latter-day Saint hymnologists began producing songs and their work became a part of the second hymnal published in 1841. The most notable of these hymnologists was W. W. Phelps. The most popular hymn written by Phelps was entitled, "The Earth Was Once a Garden Place." Commonly called "Adam-ondi-Ahman," This hymn was sung by early Saints more frequently than any other (Poulter 1995). Adam-ondi-Ahman, was according to Smith, the Garden of Eden, and its location was in Jackson County, Missouri (D&C 116). "Adam-ondi-Ahman" proclaims that the earth prior to the fall was good and filled with peace. Additionally, it declares that the prophet Enoch and his people turned from worshiping "mammon" and that the peace and beauty of the Garden was restored within the city of Enoch. The final verse is the prophetic image of an earth once more restored to her "glorious bloom," or paradisiacal glory. Phelps articulated his belief in the impending millennium and supported Smith's belief that man's actions brought about the stain upon the earth and it is man who must work to revitalize earth's original purity.

Phelps previously wrote of the land of Missouri as the gathering place of the Saints. He penned, "we cannot help exclaiming with the prophet, O land be glad! and O earth, earth, earth, hear the word of the Lord: For Zion's sake will I not hold my peace, and for Jerusalem's sake I will not rest" (Phelps 1832b).

Many of the principles taught in the early church continue today. In an official church magazine in 2019, the Church released a short article in which it outlined the parallels between the earth's existence and the stages of humanity. It read, "We all experience birth, life, death, and resurrection—and so does the earth" (Liahona 2019). The earth would evolve through three distinct states. First, the earth was created in a Paradisiacal State.

Bruce R. McConkie of the Quorum of the Twelve wrote in 1982 that "This first temporal creation of all things . . . was paradisiacal in nature" (McConkie 1982). Immediately after the Creation, nothing was mortal or subject to death (Millet 1994). Because of the transgression of Adam and Eve, the earth and humankind were in a fallen state. James E. Talmage of the Quorum of the Twelve taught that "the earth itself fell under the curse incident to the fall of [Adam and Eve], and . . . even as man shall be redeemed so shall the earth be regenerated" (Talmage 1990). Finally, the earth and humankind will be Sanctified. Russell M. Nelson, then serving as a member of the Twelve taught, "At the Second Coming of the Lord, the earth will be . . . returned to its paradisiacal state and be made new" (Nelson 2000). After the Millennium, the earth "shall be sanctified; yea, notwithstanding it shall die, it shall be quickened again," and the righteous shall inherit the celestial kingdom" (the sanctified earth; see D&C 88:17–26). The earth and humanity share a symbiotic relationship that includes theological connections. The earth and humanity share a common path, and the earth's destiny is shared with and parallel to the path of humanity.

This passage implies that the earth is alive and has a distinct and sentient spirit. Although the text can also be read symbolically and metaphorically, and need not be interpreted literally, many early Latter-day Saint leaders spoke of the salvation of the earth in terms that correspond to that of humanity (Hoskisson and Smoot 2016). Orson Pratt described the "first birth of our creation" as the earth itself being "called forth from the womb of liquid elements" and later "clothed upon with vegetable and animal existence" (Orson Pratt, 1852). In Latter-day Saint theology, the earth itself is required to receive the same rites, sacraments, and ordinances necessary for salvation. Just as each person is required to receive certain rites, so too, the earth is required to receive salvation.

In 1851, Orson Pratt interpreted the flood of Noah as a baptism (cp. 1 Peter 3:18–21). He stated that "the first ordinance instituted for the cleansing of the earth, was that of immersion in water; it was buried in the liquid element, and all things sinful upon the face of it were washed away" (Pratt 1851). Latter-day Saints primarily emerged in the nineteenth century from a Protestant background (Holland 2011). Protestants stopped short of labeling the flood a literal ordinance; though they thought of the flood as accomplishing the same end for the earth that baptism does for mortals, Latter-day Saints leaders pushed this idea much further. Latter-day Saints were "much more invested than Protestants in interpreting the Flood as a literal ordinance" (Hoskisson and Smoot 2016). W. W. Phelps, in *The Evening and the Morning Star,* previously wrote in 1832 that "the earth was washed of its wickedness by the flood; and the Son of God came into the world to redeem it from the fall." (Phelps 1832a). He later wrote in 1835 that "the earth had been baptized by a flood, for a remission of her sins." (Phelps 1835). Many Latter-day Saint writers have spoken of the baptism of the earth as a literal ordinance and one pertaining to the earth's own eternal destiny (Taylor 1884; Whitney 1885; Widtsoe 1960; McConkie 1960; Callister 2000). Additional ordinances are required for the earth, including baptism by fire (Young 1860b). Each ordinance the earth experienced is connected to the saving grace of Jesus Christ. Joseph Fielding Smith later taught that "the earth, as a living body, will have to die and be resurrected, for it, too, has been redeemed by the blood of Jesus Christ" (Smith 1954). Given as part of a revelation to the Prophet Joseph Smith in December 1832, D&C 88:26 suggests that the earth will be resurrected. The text reads, "[The earth] shall die, it shall be quickened again." The concept that the earth is a living entity needing salvific ordinances appears to "be the cornerstone of the Latter-day Saint belief" (Hoskisson and Smoot 2016).

As Brigham Young explained, "The earth is very good in and of itself and has abided a celestial law; consequently, we should not despise it, nor desire to leave it, but rather desire and strive to obey the same law that the earth abides" (Young 1855). Latter-day Saint theology consistently teaches concerning the spiritual nature of the earth, including her ultimate salvation. The relationship mankind has with her has direct ramifications concerning the eternal destiny and salvation of mankind.

## 7. View of Stewardship

God, teaching through delegation, gives responsibilities to not only care for one another but to care for this world. Delegated responsibility is given to humanity for the purpose of acting on God's behalf in the act of preserving and blessing life for the whole creation. This responsibility connects the human community both to God, to others, and by extension, to the land.

Speaking of this divinely delegated authority, Smith warned in 1839, "to exercise control or dominion or compulsion upon the souls of the children of men, in any degree of unrighteousness, behold, the heavens withdraw themselves; the Spirit of the Lord is grieved" (D&C 121:37). Humanity's responsibility to nurture creation must be viewed in the light of divinely appointed delegation. Brigham Young stated, "The very object of our existence here is to handle the temporal elements of this world and subdue the earth, multiplying those organisms of plants and animals God has designed shall dwell upon it" (Young 1862).

The book of James declares, "Pure religion and undefiled before God and the Father is this, To visit the fatherless and widows in their affliction, and to keep himself unspotted from the world" (James 1:27). Smith was deeply concerned about the poor. Flushed with convert immigrants wherever he went, caring for those in need was constantly on his mind. When faced with continual economic troubles himself, his mind often turned to possible sources of relief. In an 1834 revelation, Smith declared, "For the earth is full, and there is enough and to spare; yea, I prepared all things, and have given unto the children of men to be agents unto themselves" D&C 104:17. This promise of material bounty was couched in the context of assisting the poor and the use of self-restraint. George Handley wrote, "the promise of sufficient natural resources is only realized if we learn to consecrate our blessings for others and live with appropriate moderation" (Handley 2020). The law of consecration places emphasis not only on selfless sacrifice for others but also on the vigilant and equitable distribution of natural resources. Marcus B. Nash wrote: "LDS doctrine is clear: all humankind are stewards over this earth and its bounty-not owners-and will be accountable to God for what we do with regard to His creation. . . . how we care for the earth, how we utilize and share in its bounty, and how we treat all life that has been provided for our benefit and use is part of our test in mortality . . . The unbridled, voracious consumer is not consistent with God's plan of happiness, which calls for humility, gratitude, and mutual respect" (Nash 2013). This suggests that humanity is given an appointed responsibility concerning the earth, a theologically motivated responsibility. The divinely delegated charge extends far beyond a few environmental issues. The belief intertwines relationships with God, man, and the earth itself.

## 8. Conclusions

Smith introduced and elaborated on various elements concerning the past and the future of the planet. The prophecy of Enoch personified the earth giving her qualities that humanized her care for and pain from humanity. These all work together to illustrate the theological responsibility of all humankind. Latter-day Saint theology teaches that the creation involved not just a singular entity, but rather was a family affair, as well as that the creation is as ongoing as revelation, and that the responsibility to care for and continue to improve the environment has not ceased. The actions of individuals toward other people affect the earth in remarkable ways.

Since the early days of The Church of Jesus Christ of Latter-day Saints, caring for one's fellow human beings and caring for personal stewardship and resources have been interrelated beliefs. Smith introduced a personified earth, an earth with a soul and with an eternal destiny. The message suggests that those who understand and apply these principles of caring for others and their surroundings are welcome to make this earth their eternal home. Building on this principle, Joseph F. Smith said that "men cannot worship the Creator and look with careless indifference upon his creatures . . . Love of nature is akin to the love of God; the two are inseparable" (Kelson 1999).

**Funding:** This research received no external funding.

**Institutional Review Board Statement:** The study was conducted according to the guidelines of the Declaration of Helsinki, and approved by the Institutional Review Board.

**Informed Consent Statement:** Not applicable.

**Conflicts of Interest:** The authors declare no conflict of interest.

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
