# Peer review of "The Earth as “Mother of Men” in Latter-Day Saint Theology"

_religions, doi:10.3390/rel12111016_

Round 1

Reviewer 1 Report

This paper is a fair survey of recent work on an environmental reading of the Book of Mormon. While it is not terribly original, it does offer a nice survey and collection of a number of ideas percolating among various LDS writers and theologians. I have two primary thoughts. The first is that the author tends to make sweeping claims about LDS theology - "Latter-day Saint theology states X, or argues Y."  I don't believe LDS theology is as univocal and consistent as the author claims; rather, I think there are multiple possibilities within LDS theology, and I would prefer the author offer readings from LDS texts, or cite particular LDS authors.  The author should be claiming that LDS theology can be what the author argues, not necessarily that it *is* that already. After all, both Joseph Smith and Brigham Young made a number of claims that contemporary Saints would not say are "LDS theology" today.  Second, there are a number of small textual issues in the piece; the author uses "Latter-Day" and "Latter-day" interchangably, there are some typos, and so on. At one point the author uses the phrase "creation care," which has been coined by contemporary evangelicals - I find it interesting the author is bringing that idea into LDS theology, but would like to see some engagement with evangelical thinkers to see if LDS thought might differ.

Author Response

I am grateful for your feedback. I have made dramatic changes to the paper based on the insights you provided and hope the changes are sufficient. 

Reviewer 2 Report

Please see the attached document for feedback.

Author Response

I am deeply grateful for your feedback. Your insights were very thorough and insightful. I have made significant changes to the paper based on your advice and enjoyed reading the articles you suggested. As you look at my revision, you will easily see the shift in emphasis. Some sections were expanded, others reduced based on your counsel. 

Reviewer 3 Report

Review 

This article begins with the story of Enoch in Joseph Smith’s Book of Moses describes a personified female earth as a mother who weeps for the wickedness of humankind. The author argues that such a description should inform an LDS environmental and social justice ethic. The paper sets up an expectation that it will provide a scriptural analysis, but the execution often falls short substituting assertions for analysis and wandering off into other areas. 

There are a number of analytical problems here, beginning with arguments about the historical context of the texts under question. Tthe paper attempts to make arguments about an ancient literary context for the Book of Moses, drawing on apologetic literature that makes parallels with 1 Enoch. These are problematic claims and do not draw on the most recent scholarship. Givens and Hauglid’s book The Pearl of Greatest Price is relevant here, as well as Colby Townsend’s recent article, “Revisiting Joseph Smith and the Book of Enoch.” The arguments throughout the section on Creation further look for ancient literary parallels, but they also make it difficult to claim that these views represent Joseph Smith’s teachings. Much of the analysis could be improved by becoming familiar with the original manuscripts of the translations and revelations cited. Working from the canonical editions of these texts may have some benefits, but not when making historical claims. 

Further, there are a number of assertions in the paper that do not seem to have evidence, especially as it relates to harmonizing competing versions. This is a pervasive issue, but a few examples here. The paper argues that the Earth’s complaints about “filthiness” and “wickedness” refer to environmental pollution and social justice respectively, but as interrelated concerns. However, no evidence is given for this claim. Further, the “Gods” in the Book of Abraham are identified as “pre-mortal children of God.” In other cases, assertions are made about what “the Book of Mormon clearly teaches” giving a few citations but the citations do not clearly make that claim at all. Such an interpretation needs to be demonstrated, not asserted. Further, the analysis of the “Fall” in LDS teachings is not really rooted in the texts, but in later LDS readings. Some analysis of LDS interpretations of the Fall may be interesting for their views on environment, but they aren’t supported by the citations. The connections between temple and creation and abusive priests are not well-supported by the evidence put forward. The idea that “the theme of nature turning against the greed of mankind is amply attested in the Hebrew Bible” is not well-supported and no citations are given at all for the assertions about Isaiah.  

Organizationally, the paper needs a lot of work. The use of current LDS leaders feels too abbreviated and out of place in the section on creation. If one of the warrants is that LDS church leaders have offered teachings on environmentalism, that belongs in a separate section of analysis, not in the close reading of the creation stories. The same happens in the section on “the Fall,” which moves to ancient temples and Zion’s camp. Further, the essay often strays from its intended analysis of the Book of Moses as later sections of the paper seem to just make points about the earth or social justice issues from different texts. The paper then can feel like a hodgepodge of analysis drawn from scriptures, quotes from early and modern church leaders, and scholars, rather than a clear analysis of a particular issue. It veers into the sermonic at several points and relies too much on authority—often excessive quotations— rather than analysis. 

The core idea of this paper—that there are plausible environmentalist readings of LDS scripture, especially the Book of Moses—has some merit and there are certainly convincing arguments that might be made. However, the methodological approach in this paper, the use of evidence, and the organization make it not quite ready for publication. I encourage the author to continue to refine the idea and argument. 

Author Response

Thank you for your feedback, I appreciate how thorough and specific you were in giving advice for improvement. I have made dramatic changes to the paper and hope that I was able to follow all the counsel you gave. 

Round 2

Reviewer 2 Report

May I commend the author for responding to my comments - I feel that this paper has made the necessary changes and is a much stronger paper for it. I appreciated that there was more said about Heavenly Mother and felt the signposting of texts was more rigorous. Indeed, the structure of the argument was more concise, allowing the reader to be more fully engaged with the argument.

Before it can be published,  however, I suggest the author carefully proofread as there were a number of errors that still need addressing.

Again, many thanks for taking the time to consider my comments, which I hope were helpful and wishing you well in all your research endeavours.  

Author Response

I appreciate your feedback. Your comments have been constructive in seeing my blindspots in this paper. Thank you for your critical eye and insightful suggestions.

Reviewer 3 Report

This paper is a substantial revision of a previous version. The paper is framed as an analysis of the Book of Moses as representative of Joseph Smith's teachings about the earth. The author brings an environmental ethics to interpretation the images about the earth in this text and others. 

In this revision, the author rightly cuts many of the tangential points and some of the weaker arguments and analysis. Others, such as the inaccurate depiction of LDS views of the fall or the assertions about ancient temples and priests, remain practically unchanged. The author does not examine the original manuscripts of the Book of Moses, as I suggested, which offers a more complicated understanding of the gender of the divine characters than the one discussed here. Further, the author still relies on apologetic arguments about the antiquity of the Book of Moses, while at the same time representing it as a reflection of Joseph Smith's own teachings. 

However, there are two interrelated problems that the paper faces beyond just some warrentless claims. First, the paper remains highly sermonic. The tone and analysis of this paper as it stands are not yet suited to an academic theological paper. It concludes with an inspirational quote rather than the author's analysis of a specific issue. The author crosses over into dogma at various points, such as: "God often teaches through delegation. Responsibilities are often entrusted to his children to help them continue to progress and grow."

This issue is often compounded by the frequent quotation of LDS church leaders to buttress a point--not just in the concluding exhortation. Such citational practices are common in LDS sermonizing, but they do not add analytical heft. Instead, they substitute authority for analysis. For a paper on Joseph Smith and the Book of Moses, teachings from the current LDS church president, or Bruce R. McConkie, or others do not contribute to a textual or theological argument, but are substitutions for it. The analysis of the teachings of early LDS leaders like Phelps and others might add to greater context for the Book of Moses, but the analysis is not systematically pursued. Further, the use of such quotations suggest that LDS church leaders from Brigham Young to Bruce R. McConkie were staunch environmental theologians--this is a misrepresentation without further analysis. 

The organization of the paper then suffers as it is built around proof texts and selective quotes of church leaders, rather than exegetical or theological analysis. Sections often begin with quotations from the Book of Mormon, Doctrine and Covenants, or modern church leaders, rather than a closely organized paper around the Book of Moses. The relevance of the Book of Mormon and D&C is assumed to be simply harmonized, despite significant differences. For instance, in the first paragraph of the first section on "Mother Earth" a reference is given to the Book of Mosiah from the Book of Mormon which also refers to "mother earth." Both of these texts are then directly related to later LDS theology about a "Mother in Heaven." Such harmonization between dramatically difference concepts in all three cases is just asserted casually, rather than even argued. 

While this version is more tightly argued than the last, it remains problematic in my view. A paper on the view of the earth and nature in the Book of Moses, as promised in the paper's abstract and introduction, might have some good things to say, but this paper does not deliver. 

Author Response

I have made another round of substantial revisions to this paper. I am grateful for your insightful feedback and have incorporated many of your suggestions into this new version. I have attempted to tone down the sermonic tone of the paper and contextualize the quotes. I have also added further analysis of each section. I hope that the changes to the text are a sufficient improvement. Again, I appreciate your feedback and the critique you provided.

Round 3

Reviewer 3 Report

In my previous review, I identified three major concerns. First, a number of assertions and warrantless claims. Second, a sermonic tone including frequent citations of modern church leaders as evidence for the author's argument. Third, an organizational unity of the paper that begins with Smith's scriptural contributions but migrates away from this as the paper progresses. I also provided a number of edits in the manuscript itself, including bibliographic recommendations. 

This revision makes some progression the first with some additional notes and explanations, though many remain unconvincing. For instance, the idea that the earth is understood as Heavenly Mother or that Smith's Enoch text has "remarkable parallels" with 1 Enoch. With respect to the sermonic qualities, I do not see much progress. Entire sections are dedicated to buttressing support for the author's points from contemporary church leaders or even other historical LDS thinkers rather than close analysis of Smith's texts. Citations of devotional literature may be found throughout the text, including the conclusion. Finally, I don't see any real progress on the organization of the text itself. The second half of the essay often buries the analysis of the Book of Moses or Smith's treatment of Genesis, if it appears at all. Long digressions into other early LDS thinkers take up significant space, without much attention to potential differences. I had recommended more tightly argued "paper on the view of the earth and nature in the Book of Moses, as promised in the paper's abstract and introduction." However, this version still does not accomplish this goal.